# Involvement, Perception, and Understanding as Determinants for Patient–Physician Relationship and Their Association with Adherence: A Questionnaire Survey among People Living with HIV and Antiretroviral Therapy in Austria

**DOI:** 10.3390/ijerph191610314

**Published:** 2022-08-19

**Authors:** Helmut Beichler, Igor Grabovac, Birgit Leichsenring, Thomas Ernst Dorner

**Affiliations:** 1General Hospital, Nursing School, Medical University Vienna, 1090 Wien, Austria; 2Centre for Public Health, Department of Social and Preventive Medicine, Medical University of Vienna, 1090 Wien, Austria; 3Biomedical Science Communication med-info.at, 1150 Wien, Austria; 4Centre for Public Health, Department of Social and Preventive Medicine, Unite Lifestyle and Prevention, Medical University of Vienna, 1090 Wien, Austria

**Keywords:** HIV, AIDS, patient–physician relationship, adherence, ART, involvement, perception, understanding

## Abstract

Background: The relationship between patients and healthcare professionals (physicians) is the cornerstone of successful long-term antiretroviral therapy for people living with human immunodeficiency virus (HIV). Purpose: This study aimed to investigate the connection between involvement, perception, and understanding as the basis for the patient–physician relationship and drug adherence, measured as the probability of non-adherence. Methods: In an online survey, people with HIV were asked about their relationship with their physicians and the extent to which they felt involved in treatment-related decisions. A statistical analysis was conducted to determine whether a better patient–physician relationship was associated with higher adherence to therapy. This was performed by univariate group comparison (Mann–Whitney-U, Fishers Exact Test) and logistic regression. Results: A total of 303 persons living with HIV participated in the survey, and 257 patients were included in the analysis. Overall, 27.6% were classified as non-adherent and self-reporting based on whether an antiretroviral therapy (ART) was taken in the past or how often the ART was interrupted. This proportion was significantly higher among patients aged 50–74 years (39.7%) and those with a longer therapy duration (9–15 years: 46.6%; from 15 years on: 55.8%). Therapy-non-adherent patients showed significantly lower scores in the relationship aspects understanding (2.68 vs. 3.03), participation (2.63 vs. 3.07), and perception (3.00 vs. 3.24) compared to adherent patients. Logistic regression analysis confirms that higher scores for understanding, involvement, and perception are strongly associated with a reduction in the risk of becoming non-adherent. This was true for all examined regression models, regardless of whether they were adjusted for the length of therapy and socio-demographic characteristics. Conclusion: The results reinforce the need for awareness among health care professionals (HCP) regarding understanding, involvement, and perception as important aspects to improve the quality of the patient–physician relationship for high adherence levels with maximized non-adherence in ART management by PLWH.

## 1. Introduction

In 2019, approximately 25,000 people in the European Union (EU) were diagnosed with human immunodeficiency virus (HIV) [1]. Overall, this presented as a slight decrease in HIV diagnoses in the EU from 2008 to 2019, whereas some Eastern European countries reported an increase in incidence [2]. In 2019, approximately 400 new HIV infections were newly diagnosed in Austria. In terms of access to antiretroviral therapy (ART), people living with HIV (PLWH) in Austria have almost unlimited access to ART, which is covered by health insurance [3].

The most important goals of antiretroviral treatment in PLWH are to achieve maximal and sustainable suppression of the virus (HIV is undetectable), minimize resistance to ART, prevent HIV-associated morbidity and mortality, and prevent transmission (patients are untransmittable) [4,5]. Achieving these goals requires the optimization of both health and disease-related factors, as well as strengthening patient-related factors such as psychosocial resources, knowledge, shaping attitudes and beliefs about HIV and AIDS, and managing perceptions and expectations. Moreover, therapy adherence to potentially complex and lifelong therapy regimes is equally important and is affected by communication, cooperation, and the overall relationship with the treatment team [6,7]. In Austria, PLWH are cared for both in clinical settings, in HIV outpatient clinics, and in private practice by physicians with special expertise in HIV. Care is more difficult in rural areas because the care structure with specialists cannot be guaranteed without gaps [8]. PLWH are forced to travel to the nearest larger cities for regular blood draws and to organize prescriptions for ART [9]. The relationship between the physician and patient has positive effects on the motivation to achieve high adherence levels, and it is an essential pillar of treatment for PLWH in Austria) [10].

Involvement, perception, and understanding are complex concepts often used synonymously in antiretroviral medication management [11,12,13,14]. Various influencing factors play a role in reciprocal relationships between patients and healthcare professionals, visualized in the developed Figure 1. Involvement is closely related to the decision-making process and includes the expression of opinions during treatment [15]. Participation is conceptualized as involvement on the part of the patient in decision-making and includes voicing opinions about different treatment methods [16]. As such, it has been associated with more involvement in the health care system; in sharing information, feelings, and signs; and in accepting instructions from healthcare professionals (HCP) [17]. Participation and involvement are defined as cooperation between patients and physicians to achieve a higher degree of understanding of information and allow for co-decisions in the therapeutic process [18,19]. Moreover, the improvement of understanding and perception is a common goal when working with patients and includes patients in shared decision-making in all decisions concerning disease management [20]. Furthermore, PLWH can be actively engaged in discussing and deciding on aspects of antiretroviral therapy together with HCP [21].

Understanding is a mutual process that not only includes the understanding of information regarding treatment and disease management from the patient’s and HCP point of view with a focus on the importance of dosage, intake, and adherence to therapy [22].

Perception is another important aspect of the relationship between patients and healthcare professionals. On the one hand, the patient gains importance in being perceived as a full member of the treatment team during the entire treatment period. On the other hand, healthcare professionals are required to show interest and openness towards patients and participate in treatment management with full responsibility [23]. Moreover, the quality of the patient–physician relationship is determined by the communicative ability to ask about the patient’s perspective, the amount of information conveyed, and the competence of the physician in creating a satisfactory atmosphere for the patient [24]. These communication skills, including empathy, understanding, perception, involvement, expertise, and friendliness, are important components of a stable patient–physician relationship [25]. An optimal patient–healthcare professional relationship is an integral part of coping with chronic diseases and has positive effects on the development of self-management strategies and on the acceptance of the diagnosis, which in turn reduces non-adherence and promotes high adherence to the ART regimen [26,27].

Adherence is defined as the correct and regular use of therapies within the therapy goals, agreed upon between patient and physician, with poor or no adherence being the most important factor for therapy failure and resistance development [28,29]. Adherence may be particularly challenging in PLWH since long-term therapy adherence is particularly difficult when skipping doses does not immediately cause noticeable complications or side effects. For HIV therapy, studies have shown that the effect of antiretroviral therapy is highest with at least 95% adherence, even as modern medicine can build up a stable level of active substances in the serum, meaning that non-adherence therapy over several days will not lead to major consequences [30]. Nevertheless, continuous ART intake is crucial for long-term health, with few complications and a low risk of resistance to ART [31,32].

The primary aim of this study was to investigate the importance and impact of the involvement, understanding, and perception of the patient–healthcare professional relationship and the association between adherence and nonadherence to ART in an Austrian sample of PLWH. Furthermore, we aimed to assess the amount of perceived patient participation and treatment-related decisions focused on involvement, perception, and understanding and how these aspects have an effect on the non-adherence.

## 2. Methods

This was a cross-sectional multicenter study conducted via an online questionnaire that was accessible through a link distributed online by several non-governmental organizations in the field of HIV/AIDS.

### 2.1. Participants

A total of 303 participants participated in this survey. The participants were recruited via the “AIDS-Hilfen” organization (a national non-governmental organization working with PLWH) as well as different voluntary NGOs, through different campaigns on the internet and the local media. The inclusion criteria for participating in the study were as follows: 18 years or older, being HIV seropositive, and receiving ART. A total of *n* = 46 patients was excluded from the analysis due to not having met the inclusion criteria. All the participants signed an informed consent form.

### 2.2. Questionnaire

The survey was carried out as an online-based questionnaire with single and multiple-choice questions, provided in German and English for participants, whose first language was not German, with 33 questions structured in several sections about general information about HIV therapy, the patient–physician relationship (e.g., How much do you feel involved related to your HIV-treatment? How much would you like to be involved?), HIV treatment interruption (ten questions), HIV treatment switch (four questions), the tracking of HIV progression (six questions), socio-demographics (four questions), and the ART-related demographic (three questions).

Questions on relationship aspects focused on drug adherence (therapy understanding, patient involvement, and perception) were formed as statements on a four-point Likert scale on which the participants showed their agreement with the statement (e.g., My personal feeling: how willing is my HIV doctor to listen to me? How seriously does my HIV doctor take my problems into consideration? How good is my relationship with my HIV doctor and how easily can I talk to him about my HIV infection and its possible related problems? How good is the quality of my HIV doctor’s answers to my questions? The therapy understanding (first) aspect was conceptualized from questions about how well patients understood medical results, therapy side effects, and risks of non-adherence.

The second aspect was the patient’s involvement with therapy. This part consisted of questions about the extent to which the patient was able to decide on antiretroviral treatment as well as participate in discussions with the physician, and was willing to receive information about HIV/AIDS.

The third aspect covered questions on the patient’s perception of the attending physician, focused mainly on the physician’s abilities, willingness to communicate, and open-mindedness towards other lifestyles. Participants could rate all questions related to the relationship aspects on a four-point Likert scale or by choosing the option “prefer not to answer”.

### 2.3. Explanatory Variables

The main group of explanatory variables comprises the 3 relationship aspects: understanding, involvement, and perception. These variables were created by averaging the four-point Likert scale values of the questionnaire items related to the corresponding relationship aspect. Unanswered questions and questions answered by “prefer not to answer” were not considered for the averaged relationship aspect value.

In addition, further explanatory variables were included in the analysis to account for influential circumstances that might interact with the relationship aspects of non-adherence. These were length of therapy and sociodemographic characteristics. The sociodemographic characteristics consisted of sex, age, sexual orientation, educational level, and the Austrian regions where the patient received treatment.

### 2.4. Statistical Analysis

Descriptive statistics were applied for the relationship between adherence and all explanatory variables. Mean values and standard deviations were reported for relationship (involvement, understanding, and perception) aspect values, while frequencies and percentages were given for length of therapy and socio-demographic characteristics.

To examine group differences, the Mann–Whitney U test was used for relationship aspects (involvement, understanding, and perception) between adherent and non-adherent patients. Similarly, Fisher’s exact test was applied for the length of therapy and sociodemographic characteristics.

To determine which variables were associated with a higher risk for non-adherence, logistic regression analysis was conducted. Each regression model tested the effect of 1 of the 3 relationship aspect values. The models were estimated independently as the relationship aspect values are considered correlated and, therefore, may have otherwise caused problems of multicollinearity. In addition, each relationship aspect value was estimated unadjusted and adjusted for length of therapy and sociodemographic characteristics. The unadjusted model is further denoted Model I, while the adjusted model is Model II.

In addition, a reliability analysis was conducted on the 3 variables for relationship aspects to ensure that the corresponding questionnaire items can be aggregated (averaged) together. The reliability analysis was assessed by Cronbach’s Alpha. Values above a Cronbach’s Alpha of 0.8 were seen as sufficient for statistical analysis.

In general, a *p*-value below the threshold of 0.05 was considered as statistically significant for all tests. The analysis was conducted using the statistical software IBM SPSS 27.0 (Armonk, NY, USA).

### 2.5. Ethical Consideration

Ethics guidelines and compliance with the Helsinki Declaration for the conduct of the study were implemented. The participants were sent a link via Survey Monkey with the questionnaire. Before answering the questionnaire, the participants were informed about the study and gave their informed consent. The questionnaire could only be answered after confirmation of the informed consent.

## 3. Results

Of the 303 participants who took part in the survey, 257 (84.8%) were included in the analysis. Most of the participants were male (73.3%), homosexual (48.2%), aged 25–49 years (66.2%), and with a primary education level (49.0%).

Table 1 shows the results of the descriptive statistics of patient characteristics, with a group comparison between adherent and non-adherent patients. Statistical tests confirmed that the groups differed in terms of relationship aspects (understanding, involvement, and perception), length of therapy, and age.

Overall, adherent patients showed higher scores for all relationship aspects than did non-adherent patients. On a scale from 1 to 4, where 1 is weak and 4 is strong, adherent patients had mean score values of 3.03 for understanding, 3.07 for involvement, and 3.24 for perception. In comparison, non-adherent patients had mean values of 2.68 for understanding, 2.63 for involvement, and 3.00 for perception.

In addition, there were significant differences regarding the length of therapy and age of the patients. Patients with a longer duration of ART (>9 years) had a higher risk of becoming non-adherent. Similarly, it was shown that the rate of nonadherence increased with age in patients older than 50 years.

Although there was no significant difference between the sexes, proportionally more men were adherent than were women.

In the adherent group, the percentage for men was 76.9% and for women 21.1%, while in the non-adherent group, 63.9% were men and 31.9% were women. For the factors of sexual orientation, educational level, and origin, no significant difference was found between adherent and non-adherent individuals.

Regarding the length of therapy, the group comparison showed that non-adherent patients tended to spend more years on ART.

The group comparison for the relationship aspect values is shown in Figure 2. It contains the mean and 95% confidence intervals for all the relationship aspects. As indicated in Table 1, the mean relationship aspect values appear to be different among patient groups. In all three categories, adherent patients had considerably higher relationship aspect scores.

Similarly, the logistic regression confirmed that higher relationship aspect values are associated with a higher likelihood of adherence. The results of the logistic regression are shown in Table 2.

In Model I, the regression results show that all coefficients have significant *p*-values far below the 5% significance level and that the odds ratios (OR) were smaller than 1 in all cases. This indicates a strong association between higher relationship aspect values and a lower probability of non-adherence. More specifically, the magnitude of the odds ratios was about 0.5, for all three coefficients, meaning that an increase in one point of a relationship aspect reduces the risk of non-adherence by half (minus 50%).

After adjusting for therapy length and socio-demographic characteristics in Model II, the results were similar to the unadjusted model (Model I). All coefficients experienced a negative association between relationship aspect values and nonadherence. In magnitude, the coefficients also implied that with a one-point increase in the relationship aspect values, the risk of becoming non-adherent is reduced by more than 50%. However, only the relationship aspects for involvement and perception had a significant *p*-value, while understanding was considerably above the *p*-value threshold.

The reliability analysis with Cronbach’s Alpha confirmed that the aggregation (averaging) of questionnaire items to relationship aspects was suitable for statistical analysis. The Cronbach’s Alpha values were above the critical threshold of 0.8 for all relationship aspects, with 0.88 for understanding, 0.91 for involvement, and 0.94 for perception.

## 4. Discussion

This quantitative cross-sectional study conducted in Austria among 257 people with HIV on ART examined the importance of involvement, understanding, and perception in the patient–HCP relationship as well as to identify areas to improve the relationship between PLWH and healthcare professionals. Furthermore, the study aimed to investigate the association of patient–HCP relationship and the influence on adherence/non-adherence in a sample of PLWH in Austria.

We found that the relationship factors (examples for understanding, involvement, and perception: appreciation, empathy, accessibility, the assumption of responsibility in equal shares, the recognition of the patient as a full member of the treatment process, the promotion of health literacy, and a willingness to form a relationship with the patient) have a positive impact by reducing non-adherence. The results of this study clearly demonstrate the influencing factors between adherence/non-adherence and the relationship between PLWH and HCPs. The most important aspect for a stable patient–physician relationship is the outcome of an ART characterized by high adherence levels, low non-adherence, and an undetectable viral load, and consequently the reduction in the risk of spreading the virus. U = U—undetectable = untransmittable.

Adherence in the context of ART has been widely examined in different aspects.

Nevertheless, many adherence studies aimed to measure adherence levels in order to derive ART success [33]. Psychosocial and sociological aspects, as well as fields of communication for conducting a stable patient–physician relationship, are not considered.

Following the results of the study, we can also note that the quality of the patient–physician relationship is an important component in managing HIV, achieving a high degree of adherence to therapy, and integrating chronic disease into everyday life [34]. The results have a notable theoretical basis as studies have shown that the patient–physician relationship plays a decisive role in supporting the development of self-management strategies to integrate HIV infection and ART in everyday life [6,35,36,37], even though affording improvement in disease self-management is not always defined as a part of the working tasks of HCP. This is particularly important given the long-term characteristics of antiretroviral therapy, which may lead to the suppression of HIV replication and high adherence, with the effectiveness not being guaranteed without reliable intake [38]. Our study confirmed that involvement is an important factor in the relationship and treatment of PLWH. Not only does the relationship with HCP affect adherence/non-adherence, but also socioeconomic factors such as higher income, occupation, education level, and disease-related factors such as more symptoms and the duration of ART. Studies have shown regional distinctions in the impact of these different factors, which seem to play a decisive role in African countries [39].

Non-adherence over a longer period of time increases the risk of an AIDS outbreak [40,41,42]. Prevention strategies that focus solely on ABC rules (abstinence, faithfulness, and condom use) have reached their limits. Prevention of HIV has become an integral part of HIV treatment and prevention strategies to prevent its spread. ART offers the best protection against transmissions. The U = U (undetectable = untransmittable) campaign has become the gold standard for treatment and prevention [2,43].

Studies also underline the importance of comprehensive health literacy to support patients’ knowledge, and an understanding of ART is important for optimal medication management [44]. The effects of involvement, understanding, and perception are associated with the development of health literacy and the theory of shared decision-making as being the best practice in health care [20,45,46]. Our results suggest that the potential relationship between the involvement aspect of the patient–physician relationship does not consider the need to promote health literacy in PLWH [47,48]. Patients with higher health literacy have better therapy understanding, a more active role in shared decision-making, and higher adherence values [49].

Empowerment, health competence, and strengthening resilience are predictors of managing HIV and ART [47,50].

Various studies have clearly shown that the positive aspects of the patient–physician relationship/interaction, such as involvement, perception, understanding and quality of communication, provision of information enhance adherence to ART [9,51,52,53,54]. This is also seen in studies investigating dissatisfaction, which show that patients who are dissatisfied with their relationship with their treating physician are less adherent than those who are satisfied [15,55,56]. Additionally, other issues may increase adherence. For example, it is important to establish a schedule between the HCP and patients for antiretroviral treatment, in which appointments for laboratory controls, adherence checks, and the issuance of prescriptions are made [57,58]. This may also result in fewer appointments being made during treatment if patients manage their disease independently [59,60]. Otherwise, too much time left between appointments may be experienced as personal rejection [30,61]. In addition, the availability of physicians via e-mail outside of appointments is an advantage for participants during medication management [62]. It must also be considered that each physician consultation is not an isolated event; rather, patients are active agents in the patient–HCP relationship [63,64,65]. In addition, interpersonal and intrapersonal factors, such as stigmatization and discrimination, also initiated by the HCP, have a negative impact on adherence and increase the rate of nonadherence patients [66,67].

New data and knowledge in this area for a better understanding of the individual aspects of PLWH with more quantitative and qualitative research in the future is necessary. This will contribute to healthcare professionals’ awareness of the importance of a stable healthcare professional–patient relationship focused on adherence in ART. Moreover, additional research is needed in the area of the importance for relationship work as the basis for a successful treatment process between patients with chronic diseases and the physician.

Despite the importance of the study’s findings, some limitations should be acknowledged. First, the study focused exclusively on nonadherence, and other aspects of adherence were not considered. Second, the study had a small number of recruited patients. Third, the adherence and non-adherence of the participants were self-reported, and definitions of adherence were inconsistent across different studies and difficult to measure. Fourth, the results and statements were regionally limited, making the international generalizability of the results difficult. Fifth, the cross-sectional design cannot provide causal conclusions. Sixth, and lastly, it refers to data collection with self-reported measures in a questionnaire, where socially desirable answers may have contributed to further data distortion.

### Systematic Implications

We believe that the current findings have important systematic implications in the context of public health with regard to the promotion of shared decision-making, for which a stable patient–physician relationship is the basic prerequisite. Shared decision-making in therapy decisions is an essential aspect because the patient should be involved in therapy-related decisions. The patient is considered a full member of the treatment process with responsibility for his/her disease and ART. Further implications refer to the sound education of physicians, where students are sensitized in the areas of psychosocial factors, including communication, to pay attention to the patient–physician relationship. Public health as a scientific discipline works cross-sectional between medicine and population health maintenance. The prevention of disease and other adverse outcomes is the primary focus. Secondary prevention aspects aim to maintain a high level of health among people diagnosed with HIV/AIDS. This includes efforts to establish a variety of HIV programs to provide service guidelines to effectively reduce HIV prevalence, through health system support. A public health approach aims to address the health needs of PLWH in a population as well as the health of the population as a hole. This refers to defining the problem and risk factors as well as figuring out how to treat PLWH. From the perspective of public health, it should be noted that HIV in the context of the patient–physician relationship and the influence on adherence is an important topic regarding the prevention of HIV dissemination by lowering HIV prevalence.

## 5. Conclusions

In conclusion, based on the results of the study, it should be noted that the relationship between PLWH and physicians occupies an important pillar in antiretroviral treatment. For the success of ART, achieving high adherence and reducing the risk of non-adherence are the most important goals that must be pursued throughout life. The physicians should be aware that this is the only way to reduce complications; an outbreak of AIDS; a life expectancy largely equal to that of people without HIV, with a high degree of quality of life; and the maximized reduction in the risk of resistance against antiretrovriale substances. The establishment of a stable patient–physician relationship is the primary goal. Moreover, it is essential for physicians to consider the patient as a full member during treatment and to involve the patient in treatment. The responsibility for success remains equally with both the patient and the treatment team. Moreover, the results showed that patient involvement during antiretroviral therapy was associated with a lower chance of nonadherence to ART. Moreover, the results underline the importance of PLWH participation in the patient–physician relationship to reduce nonadherence. Patients with long-term ART should be regarded as an important target group for measures to increase adherence.

The results reinforce the need for awareness of HCP concerning understanding, involvement, and perception as important aspects to improve the quality of the patient–physician relationship in ART management by PLWHIV to promote high adherence levels in the long term and to avoid non-adherence as much as possible. Finally, the success of ART is also ultimately the most important contributor to the prevention of HIV/AIDS throughout society.

## Figures and Tables

**Figure 1 ijerph-19-10314-f001:**
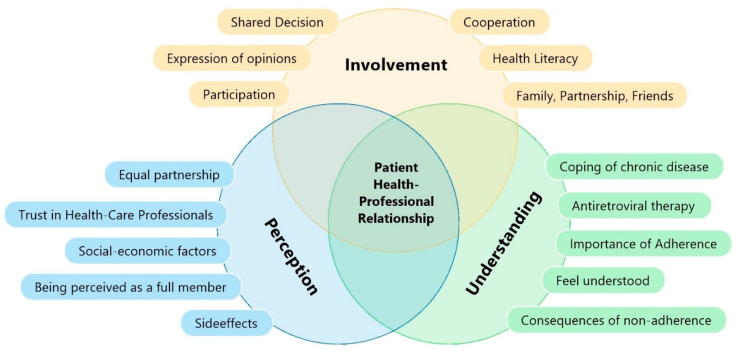
Multifactorial influences in communication on patient–healthcare professional relationship (own illustration).

**Figure 2 ijerph-19-10314-f002:**
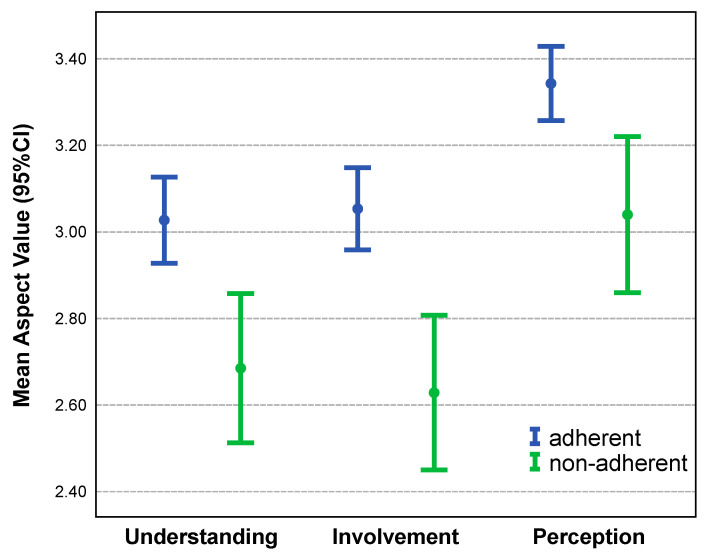
Mean aspect values (incl. 95% CI) by adherent and non-adherent patients measured on a 1–4 Likert scale (1 = weak, 4 = very strong).

**Table 1 ijerph-19-10314-t001:** Descriptive statistics on sample characteristics and inference statistics for group comparison between adherent and non-adherent patients.

	Total (*N* = 257)		Adherent (*N* = 185)		Non-Adherent (*N* = 72)		*p* *
Understanding, Mean (SD)	2.93	(0.72)	3.03	(0.69)	2.68	(0.73)	<0.001
Involvement, Mean (SD)	2.94	(0.70)	3.07	(0.64)	2.63	(0.75)	<0.001
Perception, Mean (SD)	3.24	(0.69)	3.24	(0.61)	3.00	(0.81)	0.002
Length of Therapy, *N* (%)							<0.001
Less than 1 year	19	7.4%	17	9.2%	2	2.8%	
0–3 years	55	21.4%	51	27.6%	4	5.6%	
4–8 years	80	31.1%	66	35.7%	14	19.4%	
9–15 years	60	23.3%	32	17.3%	28	38.9%	
Above 15 years	43	16.7%	19	10.3%	24	33.3%	
Gender, *N* (%)							0.065
Male	189	(73.3%)	143	(76.9%)	46	(63.9%)	
Female	62	(24.1%)	39	(21.1%)	23	(31.9%)	
Transgender	4	(1.6%)	2	(1.1%)	2	(2.8%)	
Missing	2	(0.8%)	1	(0.5%)	1	(1.4%)	
Age, *N* (%)							0.003
≤34	47	(18.3%)	42	(22.6%)	5	(6.9%)	
35–49	123	(47.9%)	89	(47.8%)	34	(47.2%)	
50–65	78	(30.4%)	47	(25.3%)	31	(43.1%)	
≥66	7	(2.7%)	6	(3.2%)	1	(1.4%)	
Missing	2	(0.8%)	1	(0.5%)	1	(1.4%)	
Sexual Orientation, *N* (%)							0.147
Heterosexuell	98	(38.1%)	65	(35.1%)	33	(45.8%)	
Homosexuell	124	(48.2%)	96	(51.9%)	28	(38.9%)	
Bisexuell	18	(7.0%)	12	(6.5%)	6	(8.3%)	
Missing	17	(6.6%)	12	(6.5%)	5	(6.9%)	
Education, *N* (%)							
Primary Education	126	(49.0%)	83	(44.9%)	43	(59.7%)	0.070
Secondary Education	72	(28.0%)	60	(32.4%)	12	(16.7%)	
Tertiary Education	51	(19.8%)	39	(21.1%)	12	(16.7%)	
Missing	8	(3.1%)	3	(1.6%)	5	(6.9%)	
Austrian Regions, *N* (%)							0.340
Burgenland	2	(0.8%)	1	(0.5%)	1	(1.4%)	
Kärnten (Carinthia)	30	(11.7%)	19	(10.3%)	11	(15.3%)	
Niederösterreich (Lower Austria)	1	(0.4%)	1	(0.5%)	0	(0.0%)	
Oberösterreich (Upper Austria)	11	(4.3%)	10	(5.4%)	1	(1.4%)	
Salzburg	32	(12.5%)	22	(12.0%)	10	(13.9%)	
Steiermark (Styria)	5	(2.0%)	4	(2.2%)	1	(1.4%)	
Tirol	49	(19.1%)	32	(17.4%)	17	(23.6%)	
Vorarlberg	6	(2.3%)	6	(3.3%)	0	(0.0%)	
Wien (Vienna)	109	(42.6%)	83	(45.1%)	26	(36.1%)	
Missing	11	(4.3%)	6	(3.3%)	5	(6.9%)	

*N* = number of observations, SD = standard deviation. Adherent means that a patient never interrupted HIV therapy, while non-adherent means that the patient interrupted therapy at least once in life. * *p*-values refer to Mann–Whitney-U Test for relationship aspect values and Fisher’s Exact Test for socio-demographic characteristics (missing values are not included in the statistical tests); understanding, involvement, and perception were measured on a 1-to-4 Likert scale (1 = weak, 2 = moderate, 3 = strong, and 4 = very strong).

**Table 2 ijerph-19-10314-t002:** Logistic regression for non-adherence on relationship aspects, including model adjustments for therapy duration and sociodemographic characteristics.

Model	Sub-Model	Variable	OR (95% CI)	*p*-Value	Nagelkerkes R-Quadrat
Model I	Understanding	Understanding	0.51 (0.34–0.75)	0.001	0.065
	(*N* = 256)				
	Involvement	Involvement	0.42 (0.27–0.63)	<0.001	0.102
	(*N* = 254)				
	Perception	Perception	0.50 (0.33–0.74)	0.001	0.067
	(*N* = 253)				
Model II	Understanding	Understanding	0.65 (0.37–1.15)	0.142	0.291
	(*N* = 226)	Length of Therapy		<0.001	
		Gender		0.349	
		Age		0.183	
		Sexual Orientation		0.943	
		Education		0.431	
		Austrian Region		0.167	
	Involvement	Involvement	0.47 (0.26–0.84)	0.011	0.451
	(*N* = 225)	Length of Therapy		<0.001	
		Gender		0.199	
		Age		0.178	
		Sexual Orientation		0.915	
		Education		0.585	
		Austrian Region		0.218	
	Perception	Perception	0.52 (0.30–0.89)	0.018	0.445
	(*N* = 224)	Length of Therapy		<0.001	
		Gender		0.306	
		Age		0.148	
		Sexual Orientation		0.876	
		Education		0.498	
		Austrian Region		0.108	

Model I = unadjusted model for relationship aspects. Model II = adjusted model for length of therapy, age, gender, sexual orientation education, and Austrian regions. The number of observations differs among models because observations were excluded if explanatory values were missing. Note: odds-ratio (OR) values are only stated for relationship aspects (understanding, involvement, and perception).

## Data Availability

The data presented in this study are available on request from the corresponding author.

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
