# Peer review of "Involvement, Perception, and Understanding as Determinants for Patient–Physician Relationship and Their Association with Adherence: A Questionnaire Survey among People Living with HIV and Antiretroviral Therapy in Austria"

_ijerph, 2022, doi:10.3390/ijerph191610314_

Round 1

Reviewer 1 Report

The Association of patient involvement, perception and understanding with adherence in people living with HIV. A quantitative questionnaire study of PLWHIV in Austria

This is an interesting contribution on the understanding of PLWHIV in Austria. Nevertheless, some changes would improve its overall likelihood of being published:

1.     The title lacks clarity. I suggest authors to rephrase the title.

2.     Please adhere to the journal’s reference system.

3.     Line 44 – authors should provide a brief explanation of this trend.

4.     Objectives, especially specific objectives, should be better described.

5.     Lines 112-114 – authors must better describe all procedures involved in data selection, particularly, online and ethical procedures.

6.     Authors must further develop the “questionnaires” section. Please include examples of items, reliability information, etc.

7.     I suggest authors to include a correlational matrix of all variables under study.

8.     Please include a section with systematic implications of your results. Specifically, public health considerations, patient-doctor relationship, and HIV stigma and stigmatization in Austria.

Best wishes.

Author Response

Reviewer 1

We thank the reviewer for carefully reading and appreciating our manuscript.

Additionally, we thank the reviewers for their many helpful, constructive suggestions to improve this manuscript.

Comment

Response

Location

The title lacks clarity. I suggest authors to rephrase the title.

The title has been changed to provide more clarity. The previous title was: “Association of patient involvement, perception and understanding with adherence in people living with HIV. A quantitative questionnaire study of PLWHIV in Austria”. Now it reads: “Involvement, perception, and understanding as determinants for patient-physician relationship and their association with adherence: A questionnaire survey among people living with HIV and antiretroviral therapy in Austria.

Titlepage

Please adhere to the journal’s reference system.

Journal`s reference system has been adjusted

Whole manuscript

Objectives, especially specific objectives, should be better described.

Objectives were presented more clearly. The primary aim of this study was to investigate the importance and impact of involvement, understanding, and perception of the patient-healthcare professional relationship and the association between adherence and nonadherence to ART in an Austrian sample of PLWH. Furthermore, we aimed to assess the amount of perceived patient participation and treatment-related decisions focused on involvement, perception, and understanding and how these aspects have an effect on the non-adherence.”

p. 3

Lines 112-114 – authors must better describe all procedures involved in data selection, particularly, online, and ethical procedures.

Thank you very much for this comment. Data selection, ethical procedures have been described. “Ethics guidelines and compliance with the Helsinki Declaration for the conduct of the study were implemented. The participants were sent a link via Survey Monkey with the questionnaire. Before answering the questionnaire, the participants were informed about the study and gave their Informed Consent. The questionnaire could only be answered after confirmation of the Informed Consent. “

p. 3-4

p. 5

Authors must further develop the “questionnaires” section. Please include examples of items, reliability information

Section “questionnaire” was added as a heading. Relevant examples of different items of the questionnaire were inserted.

p. 4

Please include a section with systematic implications of your results. Specifically, public health considerations, patient-doctor relationship, and HIV stigma and stigmatization in Austria.

Thank you very much for the good hint. A section with systematic implications regarding public health was included. “Systematic implications in the context of public health relate to the promotion of shared decision making, for which a stable patient-physician relationship is the basic prerequisite. Shard decision making in therapy decisions is an essential aspect because the patient should be involved in a therapy decision. The patient is considered a full member of the treatment process with responsibility for his or her disease and ART. Further implications refer to a sound education of physicians, where students are sensitized in the areas of psychosocial factors including communication to pay attention to the patient-physician relationship.”

“Public health as a scientific discipline works as a cross-cutting marty between medicine and population health maintenance. The prevention of disease and of other adverse outcomes is the primary focus. Secondary prevention aspects aim to maintain a high level of health among people diagnosed with HIV/AIDS. This includes efforts to establish a variety of HIV programs to provide service guidelines to effectively reduce HIV prevalence, through health system support. A public health approach aims to address the health needs of PLWH in a population but also health of the population as a hole. This refers to defining the problem and risk factors but also figuring out how to treat PLWH. From the perspective of public health, it should be noted that HIV in the context of the patient-physician relationship and the influence on adherence is an important topic regarding the prevention of HIV dissemination by lowering HIV prevalence”

p. 10-11

Reviewer 2 Report

The abstract is concise and informative. However, the authors are using specific terminology that needs more clarification. For example, the word “doctors” for someone who works in the US healthcare system might have different meanings: we prefer to use physicians because someone who has a Pharm.D. or a Ph.D. is also called a doctor.

Additionally, the authors are using people living with HIV, and the correct terminology is Persons Living with HIV. They should use the abbreviation PLWH throughout the manuscript.

The methods state that “online study,” it was assumed that the authors refer to an online survey?

The authors stated that “quantitative analysis” should say statistical analysis. Furthermore, the authors failed to state the software used for this analysis and how the participants were recruited.

The authors use EU and EEA; however, they did not define these terms.

Please be consistent in using abbreviations. The authors go back and forth between abbreviations and spelling out the terminology throughout the entire manuscript.

Since the number of new HIV cases reached a plateau, the authors failed to explain the rationale for the need to conduct this study.

The authors are presenting a figure; however, they failed to explain if this figure is taken from the existing literature or if they developed it. If this figure was taken from the literature, please credit the authors who designed it.

One major concern about the introduction is the use of studies conducted in the US describing the relationship between physicians and PLWH. However, the authors failed to explain the relevant information in Austria. It is assumed that these studies were conducted in the US, where the healthcare system differs, and how the data apply to Austrian citizens. The manuscript could be strengthened by presenting how the PLWH interacts with physicians in Austria. How often the physician sees PLWH? Do they receive a phone call reminder? Are these physicians specialized in infectious disease management, or are they GP?

Methods: The terminology is ambiguous. For example, the study states “multicenter study,” and they failed to state which centers they used. Additionally, the primary concern of this study is the lack of description of the survey used. For example, did the authors use a validated questionnaire? If so, they should reference it. If not, they must describe the questionnaire and provide it as an appendix. Finally, the study states that the survey was in German and English; however, they failed to explain why they used English.

The authors present the covariates; however, they should read a few more about how to structure a survey manuscript. Finally, some good studies used the survey as the primary methodology.

·        (Agel J, Rockwood T, Klossner D. "Collegiate ACL Injury Rates Across 15 Sports: National Collegiate Athletic Association Injury Surveillance System Data Update (2004-2005 Through 2012-2013)," Clinical Journal of Sport Medicine, Nov 2016

  • Johnson PJ, Jou J, Rhee TG, Rockwood TH, Upchurch DM. "Complementary health approaches for health and wellness in midlife and older US adults," Maturitas, July 2016
  • Blue CM, Rockwood T, Riggs S. "Minnesota dentists? attitudes toward the dental therapist workforce model," Healthcare, June 2015)

The results section has significant flaws. The authors failed to state what statistical software was used. Furthermore, in the methods, they mentioned Crown Bach Alfa and did not present it in the results. Please refer to the above-suggested studies to organize your manuscript.

Furthermore, ethical approval was at the end of the results. Therefore, this part should be in the methodology.

The discussion mentions various studies; however, the authors do not make an argument what the relevance and uniqueness of their results are. To improve their manuscript, the authors must state the novelty of their results and contribution to the literature.

The study failed to state the limitations and the need for future research.

The conclusion is informative; however, various sentences do not make sense.“It might be a common potential misjudgement of  PLWHIV, as well as of their HCP, to assume that a long-lasting experience with ART leads 342 to a constantly sustainable adherence.”

Author Response

Reviewer 2

We thank the reviewer for carefully reading and appreciating our manuscript.

Additionally. We thank the reviewers for their many helpful, constructive suggestions to improve this manuscript.

Comments

Response

Location

The abstract is concise and informative. However, the authors are using specific terminology that needs more clarification. For example, the word “doctors” for someone who works in the US healthcare system might have different meanings: we prefer to use physicians because someone who has a Pharm.D. or a Ph.D. is also called a doctor.

Thank you for bringing this to our attention: The term "doctor" was changed to “physician”.

p. 1-14

Additionally, the authors are using people living with HIV, and the correct terminology is Persons Living with HIV. They should use the abbreviation PLWH throughout the manuscript.

Thank you for this for this note especially for a neutral and non-judgmental language: “People living with HIV” changed to “persons living with HIV”.

We changed the Abbreviation “PLWHIV” to “PLWH”, accordingly.

Whole manuscript

The methods state that “online study,” it was assumed that the authors refer to an online survey?

The term “online survey” instead of “online study” is now used throughout the manuscript.

Whole manuscript

The authors stated that “quantitative analysis” should say statistical analysis. Furthermore, the authors failed to state the software used for this analysis and how the participants were recruited.

We have changed the term “quantitative analysis” to “statistical analysis”. The used software for the analysis was added. The recruiting of the participants is described in the paper.

“The analysis was conducted using the statistical software IBM SPSS 27.0.”

Section “participants” p. 3

The authors use EU and EEA; however, they did not define these terms.

Thank you very much for this hint: The terms EU/EEA are now changed to “European Union”, and it is now defined.

Introduction, p. 1-2

Since the number of new HIV cases reached a plateau, the authors failed to explain the rationale for the need to conduct this study.

Thank you for this comment. Indeed, as most countries in Europe, Austria has also reached an epidemiological plateau, which is good on the count of the incidence not growing, but it also means that there is failure to reduce the numbers. As such, the epidemiological plateau is an important starting point to consider how good therapy adherence can promote low viral loads and lead to inability to transmit the virus further. In light of this, our study examined how aspects of doctor-patient relationship play an important part. Also, beyond the mere influence on epidemiological patterns, the interest in this study was based on the question of appropriate maintenance of therapy adherence for life-long therapy and what characteristics of physician-patient relationship are associated with it.

p. 1-3

The authors are presenting a figure; however, they failed to explain if this figure is taken from the existing literature or if they developed it. If this figure was taken from the literature, please credit the authors who designed it.

The figure was developed by the authors; this has been added to the text.

p. 3

One major concern about the introduction is the use of studies conducted in the US describing the relationship between physicians and PLWH. However, the authors failed to explain the relevant information in Austria. It is assumed that these studies were conducted in the US, where the healthcare system differs, and how the data apply to Austrian citizens. The manuscript could be strengthened by presenting how the PLWH interacts with physicians in Austria. How often the physician sees PLWH? Do they receive a phone call reminder? Are these physicians specialized in infectious disease management, or are they GP?

We would like to thank for this was very helpful comment. We now reflected the literature again: The literature in the introduction refers mostly to the USA, because that is where most of studies with this issue are performed, and this is also a good argument for the necessity of our survey. We agree that the US health care system is different to the Austrian, but the study also deals with aspects not relevant to the health care system. Aspects on adherence and ART including influencing factors basically do not differ internationally. Additionally, we included information about the relationship between the Austrian physicians treating PLWH and their clients.  In Austria, PLWH are treated either in the ambulant intramural setting ("out-patient clinics in hospitals") or in the extramural setting (GPs specialized in the treatment of persons living with HIV (PLWH).

p. 1-3

Methods: The terminology is ambiguous. For example, the study states, “multicenter study,” and they failed to state which centers they used. Additionally, the primary concern of this study is the lack of description of the survey used. For example, did the authors use a validated questionnaire? If so, they should reference it. If not, they must describe the questionnaire and provide it as an appendix. Finally, the study states that the survey was in German and English; however, they failed to explain why they used English.

The questionnaire was self-made. The questionnaire does not refer to any specific source. The questionnaire exists in German and English. A small percentage of the clientele are international patients. This was added to the manuscript.

As suggested, we now added the questionnaire as an appendix.

p. 4

The results section has significant flaws. The authors failed to state what statistical software was used. Furthermore, in the methods, they mentioned Crown Bach Alfa and did not present it in the results. Please refer to the above-suggested studies to organize your manuscript.

Thank you for this important hint. The reliability analysis with Cronbach's alpha confirmed that the aggregation (averaging) of question groups to relationship aspects was suitable for statistical analysis. The Cronbach's alpha values were 0.88 for understanding, 0.91 for involvement and 0.94 for perception. These values were added to the results section.

Thank you for your support with the literature provided.

p. 5-6

Furthermore, ethical approval was at the end of the results. Therefore, this part should be in the methodology.

Ethical approval has been positioned to the methods section.

p. 5

The discussion mentions various studies; however, the authors do not make an argument what the relevance and uniqueness of their results are. To improve their manuscript, the authors must state the novelty of their results and contribution to the literature.

Thank you very much for this helpful comment. The following paragraphs were revised in the discussion section:

“This quantitative cross-sectional study conducted in Austria among 257 people with HIV on ART examined the importance of involvement, understanding, and perception in patient-HCP relationship as well as to identify areas to improve the relationship between PLWH and health professionals. Furthermore, the study aimed to investigate the association of patient-HCP-relationship and adherence in a sample of PLWH in Austria.

We found that the relationship factors (examples for understanding, involvement, perception: appreciation, empathy, accessibility, assumption of responsibility in equal shares, recognition of the patient as a full member of the treatment process, promotion of health literacy, willingness to form a relationship with the patient) have a positive impact by reducing non-adherence. The results of this study clearly demonstrate the influencing factors between adherence/non-adherence and the relationship between PLWH and HCPs. The most important aspect for a stable patient-physician relationship is the outcome of an ART characterized by high adherence levels, low non-adherence as well as undetectable viral load and consequently the reduction of the risk of spreading the virus. U=U – undetectable = untransmittable).

Adherence in the context of ART has been widely examined in different aspects.

Nevertheless, many adherence studies are aimed to measure adherence levels in order to derive ART success. Psychosocial and sociological aspects as well as fields of communication for conducting a stable patient-physician relationship are not considered.”

Furthermore, the argumentation related to the relevance and uniqueness of the results as well as the novelty and contribution to the literature was added in the discussion.

Discussion

p. 10-12

The study failed to state the limitations and the need for future research.

The limitations of the study are more clearly added. The need for future research is described.

“Despite the importance of the study’s findings, some limitations should be acknowledged. First, the study focused exclusively on nonadherence, and other aspects of adherence were not considered. Second, the study had a small number of recruited participants. Third, the adherence and non-adherence of the participants were self-reported, and definitions of adherence were inconsistent across different studies and difficult to measure. Fourth, the results and statements were regionally limited, making the international generalizability of the results difficult. Fifth limitation is the cross-sectional design, which cannot provide causal conclusions. Sixth and lastly, it refers to data collection with self-reported measures in a questionnaire, where socially desirable answers may have contributed to further data distortion.”

New data and knowledge in this area for a better understanding of the individual aspects of PLWH with more quantitative and qualitative research in the future is necessary. This will contribute to health professionals’ awareness of the importance of a stable health professional-patient relationship focused on adherence in ART. Moreover, additional research is needed in the area of the importance for relationship work as the basis for a successful treatment process between patients with chronic diseases and the physician.”

p. 12

The conclusion is informative; however, various sentences do not make sense. “It might be a common potential misjudgement of PLWHIV, as well as of their HCP, to assume that a long-lasting experience with ART leads 342 to a constantly sustainable adherence.”

Thank you for this helpful comment. The conclusion has been revised and formulated more clearly:

“In conclusion, based on the results of the study, it should be noted that the relationship between PLWH and physician occupies an important pillar in the antiretroviral treatment. For the success of ART, achieving high adherence as well as reducing the risk of non-adherence is the most important goal that must be pursued throughout life. The physicians should be aware that this is the only way to reduce complications, outbreak of AIDS, a life expectancy largely equal to that of people without HIV, with a high degree of quality of life, and the maximized reduction of the risk of resistance against antiretrovriale substances. The establishment of a stable patient-physician relationship is the primary important goal. Moreover, it is essential for physicians to consider the patient as a full member during treatment and to involve the patient in the treatment. The responsibility for success remains equally with both the patient and the treatment team. Moreover, the results showed that patient involvement during antiretroviral therapy was associated with a lower chance of nonadherence to ART. Moreover, the results underline the importance of PLWH participation in the patient-physician relationship to reduce nonadherence. Patients with long-term ART should be regarded as an important target group for measures to increase adherence.

The results reinforce the need for awareness of HCP about understanding, involvement, and perception as important aspects to improve the quality of the patient physician-relationship in ART management by PLWHIV to improve high adherence levels for a long time and to avoid non-adherence as good as possible. Finally, the success of ART is also ultimately the most important contributor to the prevention of HIV/AIDS throughout society.”

p. 11

Conclusion

Reviewer 3 Report

The authors of the manuscript investigated the influence of various factors on adherence to antiretroviral therapy. The manuscript, although well written, lacked statistical strength. A small number of recruited patients is noteworthy - only 257, and only 5 questions were devoted to the analysis of the most important part of this study, i.e. (undearstanding, involvement, perception) in the survey itself. I have great doubts as to whether far-reaching statistical conclusions can be drawn on the basis of how few questions were asked of such a small number of patients.

Due to such a small number of participants, we observe disturbances. The authors write in the text that patients over 50 years of age have lower adherence to antiretroviral therapy (line 201), but this is not true for the 66+ group.

Why in model II there is pairing with length of therapy nothing more, why was the length of therapy chosen as the next factor to analize and not, for example, also a significant age? In my opinion, the model II could be completely abandoned.

Line 288 and 289 U = U stands for undetectable = untransmittable not untransmittable = untransmittable.

Author Response

Reviewer 3

We thank the reviewer for carefully reading and appreciating our manuscript.

Additionally, we thank the reviewers for their many helpful, constructive suggestions to improve this manuscript.

Comments

Response

Location

The authors of the manuscript investigated the influence of various factors on adherence to antiretroviral therapy. The manuscript, although well written, lacked statistical strength. A small number of recruited patients is noteworthy - only 257, and only 5 questions were devoted to the analysis of the most important part of this study, i.e. (understanding, involvement, perception) in the survey itself. I have great doubts as to whether far-reaching statistical conclusions can be drawn on the basis of how few questions were asked of such a small number of patients.

Thank you for this important comment. The used questionnaire (now available as appendix) consisted of 33 questions from 7 domains related to ART and adherence. Per question, a total of 6 different additional questions were formulated. These questions were rated by the participants using the Likert scale. So, in total there were 42 questions for our analysis.

“(e.g., My personal feeling: how willing is my HIV doctor to listen to me? How seriously does my HIV doctor take my problems into consideration? How good is my relationship with my HIV doctor and how easily can I talk to him about my HIV infection and its possible related problems? How good is the quality of my HIV doctor`s answers to my questions?

Indeed, the number of included patients was lower than desired by the study team, which is a limitation of the survey. We added a corresponding sentence in the limitation section in the discussion: “Furthermore, the study had a small number of recruited patients”.

p. 4

Due to such a small number of participants, we observe disturbances. The authors write in the text that patients over 50 years of age have lower adherence to antiretroviral therapy (line 201), but this is not true for the 66+ group.

Thank you very much for this comment. We agree that subgroups are quite small. E.g., the subgroup of people aged 66 years and over only consisted of 7 participants, and therefore we avoided to draw strong conclusions regarding this age group and adherence in the text. However, as descriptive values, we kept the results in table 1.  

Results

p. 5

Why in model II there is pairing with length of therapy nothing more, why was the length of therapy chosen as the next factor to analyse and not, for example, also a significant age? In my opinion, the model II could be completely abandoned.

Thank you for this comment! We now have omitted model 2 from the manuscript. Additionally, the whole Results section has been completely revised.

p. 7-9

Results

Line 288 and 289 U = U stands for undetectable = untransmittable not untransmittable = untransmittable.

Thank you for having seen this mistake. The error in U=U was corrected.

p. 9

Round 2

Reviewer 1 Report

Thank you for implementing all the requested changes; they have improved the overall quality of the manuscript and it is now fit for publication.

Reviewer 2 Report

The authors addressed all my concerns.

Reviewer 3 Report

The authors have addressed all the comments in the commentary, the manuscript has therefore gained value and can be published.